



# The effect of vertical mixing on the horizontal drift of oil spills

Johannes Röhrs[1], Knut-Frode Dagestad[1], Helene Asbjørnsen[2], Tor Nordam[3], Jørgen Skancke[3], Cathleen E. Jones[4], and Camilla Brekke[5]

[1]Norwegian Meteorological Institute, Henrik Mohns Plass 1, 0313 Oslo, Norway
[2]Geophysical Institute, University of Bergen
[3]SINTEF Ocean, Trondheim, Norway
[4]Jet Propulsion Laboratory, California Institute of Technology
[5]UiT The Arctic University of Norway

**Correspondence:** J. Röhrs (johannes.rohrs@met.no)

**Abstract.** Vertical and horizontal transport mechanisms of marine oil spills are investigated using numerical model simulations. To realistically resolve the 3D-development of a spill on the ocean surface and in the water column, recently published parameterizations for the vertical mixing of oil spills are implemented in the open source trajectory framework OpenDrift[1]. These encompass the wave-entrainment of oil, two alternative formulations for the droplet size spectra, and turbulent mixing.

The performance of the integrated oil spill model is evaluated by comparing model simulations with airborne observations of an oil slick. The results show that an accurate description of a chain of physical processes, in particular vertical mixing and oil weathering, is needed to represent the horizontal spreading of the oil spill. Using ensembles of simulations of hypothetic oil spills, the general drift behavior of an oil spill during the first 10 days after initial spillage is evaluated in relation to how vertical processes control the horizontal transport. Vertical mixing of oil between the surface slick and entrained oil is identified

as a crucial component affecting the horizontal transport of oil spills. The vertical processes are shown to control differences in the drift of various types of oil and in various weather conditions.

## 1 Introduction

Oil spill modeling aims to describe the transport and fate of oil spilled at sea, whether from marine traffic, petroleum production, or other sources. A range of physical and chemical processes affect the fate and transport of the spilled oil that includes (but is

not limited to) submersion of surface oil into the water column by breaking waves (Reed et al., 1994; Tkalich and Chan, 2002), evaporation of light components, and formation of water-in-oil emulsions (Fingas, 2016; Li et al., 2017a). Emulsification often occurs during the first hours or days, significantly changing the viscosity and density of the oil.

  The horizontal transport of oil spilled at sea is largely determined by ocean currents, waves and winds. Ocean currents and the wave-induced Stokes drift vary with depth, and the wind drag is commonly assumed to only affect the surface slick (Reed

et al., 1994; Drivdal et al., 2014). Hence, modeling of the transport and fate of spilled oil requires knowledge of the vertical distribution of the submerged oil and the amount of oil at the ocean surface.

---

[1]https://github.com/opendrift/opendrift



A number of recent studies have contributed to understanding of the wave entrainment of surface oil (Reed et al., 2009; Zeinstra-Helfrich et al., 2016; Li et al., 2017c) and the associated droplet size spectra for the submerged oil (Zhao et al., 2014; Johansen et al., 2015; Li et al., 2017a). Wave entrainment algorithms have thereby been extended to include the effect of oil properties such as viscosity, density, oil-water interfacial tension, and oil slick thickness. It is clear that by moving oil from

the surface to the subsurface, the process of wave entrainment in turn affects horizontal transport. However, investigations of horizontal oil spill transport that both incorporate the recent developments in oil spill modeling for vertical processes and compare the model predictions to field experiments are rare. In this study, we aim to study the processes that govern horizontal oil spill transport and to characterize how they are affected by wave entrainment, vertical mixing, and weathering.

Elliott (1986) first described a mechanism for spreading by current shear that showed how horizontal transport is strongly

affected by the vertical distribution of oil during a spill. Their work was based on the observation that an oil slick tends to elongate in the direction of the wind, with the leading edge of oil being thicker due to the interaction between wave entrainment and current shear caused by wind and Stokes drift. Description of the droplet sizes that result from entrainment by breaking waves is a key component in oil spill modeling, first described by Delvigne and Sweeney (1988). Mackay et al. (1982) and Daling et al. (1990) provide a description of water-in-oil emulsification, which has a large impact on the submersion and

resurfacing of oil. Many of today's oil drift models are based on these works and a Lagrangian formulation for oil particle tracking (De Dominicis et al., 2016; Spaulding, 2017).

More detailed observations and parameterizations of the droplet size spectra have emerged in recent years by several groups: Johansen et al. (2015) present a droplet size distribution for natural dispersion of oil that also accounts for oil film thickness; Li et al. developed semi-empirical models for both droplet size spectra and wave-entrainment (Li et al., 2009, 2017c, b); and Li

et al. (2017a) present observations of droplet size spectra under the effect of added dispersants, including detailed descriptions of the droplet break-up process under breaking waves. All of these studies address the effects of emulsification, wave and weather conditions, and oil properties.

Comparisons of oil spill models with actual oil spills are generally sparse, and more commonly based on drifter trajectories (De Dominicis et al., 2016). Jones et al. (2016) studied the coupling between vertical and horizontal transport within the frame-

work of a model and demonstrated that modeling of the sub-surface part of an oil spill is essential to capture the development of the surface oil slick. Their model results were compared to a time series of oil slick observations made with an airborne synthetic aperture radar (SAR). The study concluded that there is a continuous exchange between the surface and entrained oil, such that the submerged oil acts as a reservoir that constantly releases oil to the surface. Another recent study that addresses interactions between vertical and horizontal transport, focusing on smaller spatial scales, is Simecek-Beatty and Lehr (2017),

investigating the effect of Langmuir circulation on oil spill spreading.

In the present study, we use the same oil drift model (OpenOil) as Jones et al. (2016), but with further developments that explicitly model the droplet size spectra and wave entrainment based on new parameterizations (Johansen et al., 2015; Li et al., 2017b, c). This removes the need to use oil droplet size and entrainment length scale as tuning parameters in the model, as was done by Jones et al. (2016).



Further potential advances in oil spill modeling result from more detailed schemes for vertical mixing of particles (Visser, 1997; Thygesen and Ådlandsvik, 2007), surface interaction (Nordam et al., 2018), and improved ocean models. These schemes account for the vertical variation of eddy diffusivity as present at the base of the mixed layer. High resolution ocean models with data assimilation schemes provide details on the vertical structure of currents, stratification, and turbulent mixing, allowing

more realistic particle transport simulations (Sperrevik et al., 2017). For instance, vertical eddy diffusivities are commonly available as output from ocean models and can be used for vertical particle diffusion (e.g., Röhrs et al., 2014), removing the need for simpler parameterizations that are based on wind speed only.

The current study reported here was undertaken to implement and test the various new parameterizations that are relevant for oil spill transport into a comprehensive oil spill model, and to describe the potential outcome of oil spills with regard to

both vertical and horizontal transport. The model OpenOil is used here, which is part of the open source trajectory modeling framework OpenDrift (Dagestad et al., 2018). Besides the new formulations for oil droplet size spectrum and wave entrainment, we also make use of a state-of-the-art description for vertical mixing (Nordam et al., 2018) and the Automated Data Inquiring for Oil Spills (ADIOS) weathering module that is available as an open source package[2] from the U.S. National Oceanic and Atmospheric Administration (NOAA). The first aim of this study is to validate the integrated oil spill model against open

ocean observations, as done in Jones et al. (2016). Secondly, we perform ensemble simulations for an example region in the Norwegian Sea to evaluate how the various vertical processes incorporated into the oil spill model interact and affect the horizontal transport.

This paper is organized as follows. In section 2, we present the physical mechanisms and document the numerical schemes and parameterizations that are used to describe vertical oil transport. In section 3, we describe the open source oil spill model

OpenOil, including its new components. A validation against open ocean observations of an oil spill is presented in section 4. Section 5 describes ensemble simulations of oil drift for an example region and presents its results. A discussion of the model results and the physical mechanisms for vertical and horizontal oil spill transport is provided in section 6, followed by concluding comments in section 7.

## 2 Mechanisms for vertical mixing of oil

During an oil spill, the liquid oil phase is distributed between a surface slick and submerged droplets at different depths, in addition to potentially stranded oil. Vertical exchange between the surface slick and various vertical layers in the ocean results from a complex interplay of (i) entrainment from breaking waves, (ii) rise and resurfacing of submerged oil due to buoyancy, and (iii) oceanic turbulent mixing (Elliott, 1986). Physical and chemical properties of the oil control the droplet sizes, buoyancy, and the ability of waves to break up the surface slick. In the following we give a brief overview of numerical

schemes and parameterizations that describe these processes and discuss the choice of methods for this study.

---

[2]https://github.com/NOAA-ORR-ERD/OilLibrary





## 2.1 Entrainment from breaking waves

The intrusion of surface slick oil into the water column has commonly been described using an entrainment rate, which is a function of the sea state and oil properties (Delvigne and Sweeney, 1988; Delvigne and Hulsen, 1994; Tkalich and Chan, 2002). Newer parameterizations of wave entrainment explicitly include the effect of viscosity, density, and oil-water interfacial tension (Reed et al., 2009; Zeinstra-Helfrich et al., 2016; Li et al., 2017c).

The sea state is described by the area fraction of the sea surface covered by breaking waves, which is often parameterized by the wind speed and a minimum wind speed at which whitecapping occurs. At present, there are a various formulations for the wave breaking fraction (e.g., Holthuijsen and Herbers, 1986; Zhao and Toba, 2001; Callaghan et al., 2008). It should be noted that there remains large uncertainty in the relationship between the wind speed and wave breaking area fraction.

Following Li et al. (2017c), the entrainment rate $Q$ is parameterized by the dimensionless Weber (We) and Ohnesorge (Oh) numbers:

$$Q = 4.604 \cdot 10^{-10} \cdot \text{We}^{1.805} \text{Oh}^{-1.023} F_{bw} \tag{1}$$

where $F_{bw}$ is the fraction of the sea surface covered by breaking waves per unit time, as given by Holthuijsen and Herbers (1986) and subsequently used in Delvigne and Sweeney (1988):

$$F_{bw} = \begin{cases} 0.032\frac{s}{m} \cdot \frac{U_{10m} - U_0}{T_p} & \text{if } U_{10m} > U_0, \\ 0 & \text{otherwise,} \end{cases} \tag{2}$$

where $U_{10m}$ is the wind speed at 10 m above the sea surface, $U_0 = 5\,\text{m/s}$, and $T_p$ is the peak (or significant) wave period. The Weber number, We, is a dimensionless number describing the relative importance of inertial forces and oil-water interfacial tension. It is a function of the sea water density, $\rho_w$, the significant wave height, $H_s$, and the oil-water interfacial tension, $\sigma_{o-w}$, and is given by

$$\text{We} = \frac{\rho_w g H_s d_o}{\sigma_{o-w}}, \tag{3}$$

where $g$ is the acceleration of gravity and $d_o$ is the Rayleigh-Taylor instability maximum diameter:

$$d_o = 4\sqrt{\frac{\sigma_{o-w}}{g(\rho_w - \rho_o)}}. \tag{4}$$

The Ohnesorge number, Oh, is a dimensionless number describing the ratio of viscous forces to inertial and surface tension forces. It is a function of the oil viscosity, $\mu_o$, oil density, $\rho_o$, and oil-water interfacial tension:

$$\text{Oh} = \frac{\mu_o}{\sqrt{(\rho_o \sigma_{o-w} d_o)}}. \tag{5}$$

The probability, $p$, for a particle at the surface to be entrained by a breaking wave and submerged is

$$p = 1 - e^{-Q\Delta t} \tag{6}$$





where $\Delta t$ is the time step in the mixing scheme. For $Q\Delta s \ll 1$, the product of wave entrainment rate and time step may be used to approximate the probability.

## 2.2  Resurfacing of submerged oil

Most petroleum products have a density lighter than sea water. The submerged oil droplets rise due to their buoyancy, which

is controlled by their droplet size and the density difference between the oil and water. This results in a terminal vertical rise velocity, $w$, that depends on the Reynolds number for the flow around the droplet, following the Stokes Law for low Reynolds numbers and an empirical expression for higher Reynolds numbers (Tkalich and Chan, 2002):

$$
w = \begin{cases} 2g\frac{(1-\frac{\rho_o}{\rho_w})}{9\nu_w}r^2, & \text{if Re} \leq 50 \\ \sqrt{\frac{16}{3}g(1-\frac{\rho_o}{\rho_w})}\sqrt{r}, & \text{if Re} > 50 \end{cases} \tag{7}
$$

where $d$ is the droplet diameter, $r = d/2$ is the droplet radius, $\nu_w$ the kinematic water viscosity and the Reynolds number, Re, is

given by $\text{Re} = 2rw/\nu_w$. Typical rise velocities range from the order of 1 cm per hour for a droplet diameter of 10 micrometers and density of 900 $kg/m^3$, to 30 meters per hour for a diameter of 500 micrometers.

## 2.3  Turbulent mixing

While buoyant forces move oil particles in one direction, oceanic turbulence entrains oil droplets both up- and downwards. Turbulence is a property of the oceanic environment and largely depends on the wind history through vertical current shear,

sea water stratification, and dissipation of wave energy. The amount of turbulent mixing is commonly described by an eddy diffusivity coefficient. The eddy diffusivity can be provided by ocean circulation models (e.g., Warner et al., 2005), or approximated by wind speed (e.g., Sundby, 1983). Using eddy diffusivities from the ocean circulation model has the advantage that it not only takes into account the local forcing from wind, but also advection and inertia of turbulence and buoyancy production and inhibition by stratification.

The vertical flux, $F_D$, of oil droplets due to turbulent diffusive transport can be described as

$$
F_D = K(z)\frac{dC(z)}{dz} \tag{8}
$$

where $K(z)$ is the eddy diffusivity and $C(z)$ is the local concentration of oil droplets. In Lagrangian particle tracking, the turbulent flux $F_D$ can be represented by a random walk process, as done by Visser (1997). In this scheme, a random vertical displacement $\Delta z$ for each particle is given by

$\Delta z = K'(z_n)\Delta t + R\sqrt{\frac{2}{r}K\left(z + K'(z_n)\Delta t/2\right)\Delta t}$                 (9)

where $K'(z)$ is the derivative of $K(z)$ in the vertical direction, evaluated at $z$, and $R$ is a random number with zero mean and standard deviation of $r$. The second term on the right-hand side accounts for vertical gradients of the eddy diffusivity,



opposing artificial offsets of random walk in the presence of diffusivity gradients. Without this correction term, particles would erroneously accumulate in regions of low eddy diffusivity (Visser, 1997; Thygesen and Ådlandsvik, 2007).

If both the terminal velocity and the eddy diffusivity are constant with depth, a steady state solution of the Eulerian advection-diffusion equation in the vertical can be found by balancing the divergence of the diffusive flux (Eq. 8) with the advective flux

$wC(z)$ due to buoyancy:

$$wC(z) + \frac{d}{dz}\left(K\frac{dC(z)}{dz}\right) = 0 \qquad (10)$$

The solution is an exponential decay of concentrations with depth,

$$C(z) = C_0 e^{-\frac{w}{K}z}, \qquad (11)$$

from a concentration of $C_0$ at the surface, which is set to a constant for this solution. This solution has been found and verified

from observations for, e.g., pelagic fish eggs (Sundby, 1983) and marine plastic (Kukulka et al., 2012). However, oil droplets can stick to the surface and the eddy diffusivity is a function of depth. Hence, the full solution of Eqs. 8 and 9 is more complex, but this steady state solution (Eq. 11) provides a general description of the vertical droplet distribution.

## 2.4 Droplet size distribution

Submerged oil droplets vary in diameter over several orders of magnitudes from $\mathcal{O}(10^{-6}\,\mathrm{m})$ to $\mathcal{O}(10^{-3}\,\mathrm{m})$. The diameter of

oil droplets affects the advective flux due to buoyancy through Eq. 7. It is known from laboratory experiments that the smallest droplets are most abundant (Delvigne and Sweeney, 1988; Li et al., 2017a). Droplet size distributions may be expressed either as a number size distribution or as a volume size distribution (Johansen et al., 2015). Although Delvigne and Sweeney (1988) observed a power-law number size distribution, recent laboratory studies indicate that the droplet size distribution is better represented by a log-normal expression (Johansen et al., 2015; Li et al., 2017b), or as two regimes with different power-law

exponents (Li et al., 2017a) that provide a separation of the droplet size spectra into surface-tension-limited and viscosity-limited regimes (Johansen et al., 2015). It should be noted that a number size distribution with two power-law regimes, as in (Li et al., 2017a), leads to a maximum in the volume size distribution such that the volume size distribution assumes similarity with a log-normal distribution.

In contrast to the power law distribution of Delvigne and Sweeney (1988) (exponential -2.3 ± 0.06), both Johansen et al.

(2015) and Li et al. (2017b) propose log-normal distributions to represent the droplet size diameters of submerged oil particles. The respective studies calculate a median droplet size based on nondimensional numbers – Reynolds number and Weber number in Johansen et al. (2015), or Ohnesorge number and Weber number in Li et al. (2017b)). The latter variants also differ in that Johansen et al. (2015) incorporate the oil film thickness into their dimensional analysis, whereas Li et al. (2017b) use the Rayleigh-Taylor instability diameter as a length scale in their droplet size spectrum, as is done for the entrainment rate (Sec.

30  2.1 ).





Both Johansen et al. (2015) and Li et al. (2017b) droplet size distributions are sensitive to oil type – a more viscous oil will have a droplet spectrum shifted towards larger droplets. An increase in wave height of the breaking waves (more energy) will shift the spectrum towards smaller droplets. The Johansen et al. (2015) distribution is also sensitive to the surface oil slick thickness, shifting the spectrum towards larger droplets with increasing slick thickness, as also confirmed by Zeinstra-Helfrich et al. (2016).

The droplet spectra from both Johansen et al. (2015) and Li et al. (2017b) are implemented in OpenOil. In section 4, we test both formulations and in section 5 we use the latter because the oil film thickness is not a state variable nor known a-priori, and may only be estimated with considerable uncertainty.

In Li et al. (2017b), the volume droplet size spectra is described by the median droplet diameter, $D_{50}^V$, as

$$D_{50}^V = d_o r \left(1 + 10\mathrm{Oh}\right)^p \cdot \mathrm{We}^q \tag{12}$$

with the empirical coefficient $r = 1.791$ and the exponents $p = 0.460$ and $q = -0.518$, using the Weber number and the Ohnesorge number from Eqs. 3 and 5.

In Johansen et al. (2015), the number size spectrum is described by the median droplet diameter for the number size distribution, $D_{50}^N$,

$$D_{50}^N = Ah\mathrm{We}^{-a} + Bh\mathrm{Re}^{-b}, \tag{13}$$

which is the sum of two terms; the first for the interfacial tension-limited regime that depends on the Weber number, We, and the second term for the viscosity-limited regime that depends on the Reynolds number, Re. Johansen et al. (2015) determined the empirical coefficients to be $A = 2.252$ and $B = 0.027 \cdot A$, and confirmed the exponents for both regimes as $a = b = 0.6$. The Reynolds number and Weber number for Eq. 13 are given by

$$\mathrm{Re} = \frac{\rho_o h \sqrt{gH}}{\mu}, \mathrm{We} = \frac{\rho_o hgH}{\sigma_{o-w}} \tag{14}$$

according to Eq. 7 from Johansen et al. (2015). H is the fall height for oil droplets in free fall (plunging) wave tank experiments, and is found to be equivalent to wave height (Reed et al., 2009).

The median diameter, $D_{50}^N$, is then transferred to the volume size median diameter using the relationship

$$\ln D_{50}^V = \ln D_{50}^N + 3(s \ln 10)^2 \tag{15}$$

given the logarithmic base-10 standard deviation which was determined to be $s = 0.38 \pm 0.05$ from the experiments.

Finally, for both formulations the probability density function (pdf) from which each droplet diameter is drawn is given by

$$\mathrm{pdf}(d) = \frac{1}{ds\sqrt{2\pi}} e^{-\frac{(\ln d - \ln D_{50}^V)^2}{(2s^2)}}. \tag{16}$$



Using volume or mass distributions instead of number distributions in Lagrangian oil spill modeling reduces the computational cost for the simulations, as a smaller number of particles is required to represent an oil spill spreading in a given region. Since the large majority of oil droplets are very small, Lagrangian particle tracking with number distribution would follow many small particles and few large particles. However, most of the oil mass is found in oil droplets of intermediate size. This is

true for the log-normal number distributions (Johansen et al., 2015; Li et al., 2017b), as well as for the power-law distribution with two separate regimes of varying exponent in (Li et al., 2017a). Using volume distribution spectra provides a justification to truncate the droplet size spectra away from its peak, such that the smallest particles with negligible mass, as well as the largest particle with negligible number can be disregarded. Following mass is also the more applicable approach for many purposes, i.e., to identify where to look for the higher concentrations of oil in cleanup operations.

## 3   Simulations with the oil drift model *OpenOil*

The presented oil drift simulations are based on the open source trajectory modeling framework OpenDrift (Dagestad et al., 2018). A sub-class of OpenDrift is OpenOil, manifesting an oil drift model that includes the specific descriptions of processes relevant for oil spills as described in section 2. Other functionality that is not specific to oil is inherited from OpenDrift, such as advection by ocean currents and processing of environmental input data. In the following we describe the environmental data

used for this study and how the mechanisms described in section 2 are implemented in OpenOil.

### 3.1   Particle advection by currents, wind, and waves

In this study, particles are advected horizontally by ocean currents provided by the Norshelf reanalysis (Christensen et al., 2017), covering the shelf sea off Norway at 2.4 km horizontal resolution and with 42 vertical layers. Norshelf is based on the Regional Ocean Modelling System ROMS (Shchepetkin and McWilliams, 2005) including a 4D-variational data analysis

scheme (Moore et al., 2011) that uses openly available observations of hydrography. The ocean currents from Norshelf are available at hourly time steps. Particles are advected using an Euler-forward scheme and a time step of 15 minutes. OpenDrift provides the flexibility to use various types of environmental input data, and for the particular example presented in section 4, ocean currents from in-situ observations are used instead of ocean model results. Wind and wave data are from the ERA-interim reanalysis provided by the European Center for Medium Range Weather Forecasts (ECMWF), which is the same atmospheric

data that is used in the ocean model Norshelf.

Oil particles at the sea surface are additionally subject to direct wind drag, using a wind drag coefficient of 0.02 (Jones et al., 2016). Note that some oil spill studies refer to a wind drag factor of 0.03 (e.g., Reed et al., 1994), which has also been inferred for drifters on the sea surface (Dagestad and Röhrs, 2018). Using a lower coefficient of 0.02 accounts for the fact that parts of the commonly observed wind drift is due to Stokes drift at the surface (Drivdal et al., 2014). The Stokes drift from surface

gravity waves is treated separately in OpenOil, and obtained from a wave forecast model. Stokes drift below the surface is calculated from the surface Stokes drift using the parameterization provided by Breivik et al. (2014).



## 3.2 Vertical transport of particles

During each time step in OpenOil (15 for this experiment), an internal loop with a time step of 60 seconds is executed for the vertical transport processes in the following order (Nordam et al., 2018):

**(i)** Turbulent mixing of submerged particles using random walk (Eq. 9)

**(ii)** Reflection of particles that hit the sea surface during (i)

**(iii)** buoyant rise of particles using their individual terminal velocity (Eq. 7)

**(iv)** set particles that reached the surface during [iii] to the exact surface. These become part of the surface slick and are no more subject to vertical mixing during subsequent time steps.

**(v)** surface particles that become subject to wave breaking, given the probability of Eq. 2, break up from the surface slick and become submerged. These particles are now subject to vertical mixing in subsequent time steps.

The eddy diffusivity coefficients for (i) are provided by the ocean model NorShelf/ROMS, based on the generic length scale mixing scheme (Umlauf and Burchard, 2005). Nordam et al. (2018) shows that the random walk for Lagrangian particles, performed as steps (i) - (iv), is equivalent to vertical mixing of Eulerian concentrations with a Neumann boundary condition enforcing no diffusive flux at the surface.

## 3.3 Wave entrainment

Step (v) depends on the description of wave-induced breakup of the surface slick. In this study the entrainment rate of Li et al. (2017c) is used (Eqs. 1 - 5). Alternatively, OpenOil also provides the option to use entrainment rates of Tkalich and Chan (2002).

## 3.4 Droplet size distribution

During wave entrainment in step (v), the oil droplet size is drawn from a log-normal random distribution (Eq. 16). The medium diameter is calculated from either Eq. 12 based on Li et al. (2017b), or Eqs. 13 and 15 based on Johansen et al. (2015).

The size of a single droplet is changed by re-drawing from a random distribution each time the particle is submerged from the surface. The particles used in the model discussed here represent super-particles composed of a variable number of droplets, but a total mass that is conserved during the wave entrainment. This has to be taken into account when interpreting the results.

To have a good statistical representation of the oil spill, it is necessary to have a large number of particles at every region of the simulation. As particles spread during the simulation, the statistical representativeness is eventually lost, setting a limit on how long the simulation can be carried out with a given number of super-particles.





## 3.5 Oil weathering and oil properties

OpenOil includes parameterizations of the oil weathering processes that are most relevant on time-scale of hours to days, i.e., evaporation and emulsification. Dispersion is not treated separately in OpenOil because it is explicitly calculated by the vertical mixing scheme in adjunction with wave breaking entrainment and droplet size distributions.

Oil properties are obtained from the open source ADIOS oil database developed by NOAA (Lehr et al., 2002). This database is accompanied by a software library (written in Python, like OpenDrift) of chemistry and weathering algorithms, which are used to calculate the evaporation and emulsification of a given oil type. The rate of evaporation varies strongly among various oil types, and depends on the wind speed. Some oil types may be completely evaporated within a few hours, whereas others do not evaporate at all. More typically, 20%-40% of the oil (the lighter components) is evaporated within the first 6-12 hours.

In addition to the removal of oil components from the sea surface, evaporation may also play an important role in the onset of emulsification, which has a very strong impact on the oil viscosity. Whereas some oils do not form water-in-oil emulsions, other oils only start to emulsify after some of the components have evaporated (Fingas, 2016, chapter 8). The water content in the emulsion may then quickly rise from 0% at onset of emulsification, to a maximum level that is typically 80%-90% for most oil types in the ADIOS database (Lehr et al., 2002). The density of the emulsion thus gradually approaches that of water, but

more importantly for transport and fate modelling, as well as cleanup operations, the viscosity may increase by several orders of magnitude.

In addition to the prediction of the effects of evaporation and emulsifications, the oil library also provides the oil-water interfacial tension, which is an important parameter for the breakup of oil into smaller droplets during entrainment events (Sec. 2.1).

## 20  4   Oil-on-water experiment in the North Sea, June 2015

For comparison with previous work, and to compare the implementation of vertical oil mixing with observations, the oil spill simulations of Jones et al. (2016) are repeated in this work using the new parameterizations for oil droplet size spectra and wave entrainment. The modeling in Jones et al. (2016) was performed with the same model as here (OpenOil), but then with constant droplet size and wave entrainment length scale as tuning parameters. The new implementation does not require a-

priori knowledge of droplet size or wave entrainment because oil properties, wind, and wave conditions are used to calculate droplet sizes and wave entrainment.

### 4.1   Experiment layout

During a measurement campaign in the northern North Sea, Jones et al. (2016) obtained synthetic aperture radar (SAR) images using the Uninhabited Aerial Vehicle SAR (UAVSAR), which mapped the surface slicks of several oil spills over the course of

7.5 hours after the initial release at the surface. We will here focus on the drift of an emulsified mixture of 40% *OSEBERG* oil, 40% *TROLL* oil and 20% sea water, which was accompanied by a GPS-tracked surface drifter. The drifter (CODE type at 0.3m



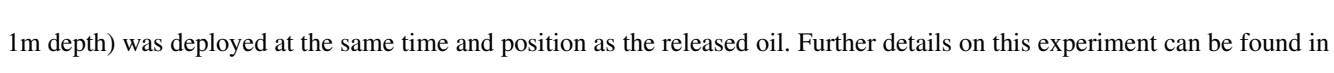

- 1m depth) was deployed at the same time and position as the released oil. Further details on this experiment can be found in Jones et al. (2016), and details on the SAR measurements of the oil slick is given in Skrunes et al. (2016).

The observed oil slick contours are shown in Fig. 1a, along with the trajectory of the CODE drifter. During an intermittent time period no images were taken because the aircraft had to refuel. In Fig. 1b-e., grayscale colors are used for the modeled
results for this period.

## 4.2   OpenOil model configuration

As in Jones et al. (2016), simulations are initiated with 10000 oil elements (super particles) seeded at the ocean surface within the area of the first SAR-observed oil slick contour at 05:48 UTC (red contour on Fig. 1a). As the evolution of a mixture of two oils is not supported by ADIOS and OpenOil, we use instead only the OSEBERG A oil type (Table 1), with 20% water content.
As the oil in the experiment was pre-emulsified before it was released on the sea, we do not calculate further evaporation and emulsification throughout these simulations, but keep the water fraction constant during the 7.5 hours.

The ocean current used for the simulations is derived from the motion of the CODE drifter, where the Stokes drift at a depth of 0.3m has been subtracted. While the Stokes drift varies with depth, the Eulerian current is assumed to be constant throughout the mixed layer. The fact that the buoy moved coherently with another buoy deployed 3 km further south (shown in Jones et al.
(2016)) indicates that the current was horizontally uniform near the released spills. These assumptions are consistent with the dynamics of inertial oscillations in a strongly mixed boundary layer as seen from the drifter motion (Qi et al., 1995).

Submerged oil elements are moved horizontally with the drifter-derived current at each time step (10 minute interval for this experiment), in addition to the Stokes drift at the depth of the given element. The depth-dependent Stokes drift is calculated from the surface Stokes drift as obtained from the WAM wave model Breivik et al. (2014). For oil elements at the ocean surface,
an additional wind drag of 2% of the wind velocity is added. Wind data is obtained from the Norwegian Meteorological Institute's operational 2.5 km AROME model. It should be noted that the magnitude and directions of both the wind and waves from the models used here corresponded very well with what was measured and estimated from the ship, and thus the environmental conditions (wind, waves and currents) are quite well known during this experiment.

For the wave entrainment, we use the parameterization of Li et al. (2017c), as described in SResponse operations require
forecasts of the oil spill position and its composition within a time frame of hours to a few days in advance. Environmental impact studies aim to identify the long-range transport of potentially spilled oil and its composition and distribution in the water column (e.g., French McCay, 2004).

ec. 2.1. For the droplet spectrum, we test both parameterizations of Li et al. (2017b) and Johansen et al. (2015) (Sec. 2.4). As the latter also depends on the surface film thickness, we roughly estimate this as 0.7 micrometers by dividing the amount of
oil (500 litres) over the area of the spill as observed from UAVSAR after 3 hours, when the spill's areal extent stabilised (0.7 km$^2$).





### 4.3 OpenOil model results

Fig. 1b shows the model results using droplet size spectra of Li et al. (2017b) and Fig. 1d shows model results using the droplet size spectrum of Johansen et al. (2015). Only the surface oil is shown here to match the observations, as the SAR images do not show the sub-surface oil.

As demonstrated in Jones et al. (2016), modeling the sub-surface oil is needed to match the observed surface slick pattern. Vertical transport of sub-surface oil continuously returns oil to the surface, revealing the position of the sub-surface oil. The surface slick is hence a result of surface oil that has been transported eastward with the wind and re-surfaced oil which has moved westward following an inertial oscillation with the ocean current. While the western tip of the oil slick is a result of the ocean current, the elongation of the surface slick towards the east is a result of wind and Stokes drift. This combination of

mechanisms, first described by Elliott (1986), is considered to be responsible for the initial spreading of oil slicks (Galt and Overstreet, R., 2011) under windy conditions.

With the Li droplet spectrum (Fig. 1b), the modeled surface slick agrees quite well with the observations. However, the elongation of the slick in the downwind direction during the first 4 hours (before the airplane refueling, red-yellow colors) is somewhat less than what is observed. This indicates that too much surface oil was entrained in the model. The entrainment

rate depends strongly on the oil viscosity, which again depends mainly on the emulsion water content. During sensitivity experiments, we increased the emulsion water content in the model to improve the match with observations. The best fit for the Li spectrum is obtained by using a water content of 40%, as seen in Fig. 1c.

With the Johansen spectrum (Fig. 1d), very little oil is found on the surface, indicating too much entrained oil, and hence too little surface drift in the downwind direction (eastwards). As with the Li spectrum, we adjust the water content to obtain the

optimal match. The best fit for the Johansen spectrum is obtained by increasing the water content to 80%, as seen in Fig. 1e.

The simulations with modified degree of emulsification, parametrized by the water content, provide an excellent fit to the observed oil slick contours of Fig. 1a, for both the first hours and the subsequent hours. There is considerable uncertainty in how much the initial water content of 20% changed after release of the pre-mixed oil, and best fits to the two available spectra yield different estimates of the oil-to-water ratio. However, an increase of the water content to 40% or 80% is very reasonable.

For many oil types, an increase of water content from 0% until saturation of about 90% within some hours is quite common (Fingas, 2016), where the speed of emulsification increases with wind and waves. Wind speed during this experiment was 9-12 m/s, allowing for a high emulsification rate.

For the comparisons it is also important to keep in mind that the SAR images do not necessarily capture the thin oil surface film on the down-wind trail. In fact, it has been reported during the measurement campaign that the SAR observations did not

show the entire oil slick that was visible from the nearby ship (Jones et al., 2016).

The new implementation of OpenOil is generally able to reproduce the oil spill transport observed during the oil-on-water experiment in Jones et al. (2016). While the previous model formulation was largely dependent on tuning of model parameters to the observations, the new model implementation can reproduce the oil slick transport qualitatively without a-priori





knowledge. However, the results still benefit from fine-tuning of the water content, illustrating that the oil weathering and emulsification is a very sensitive component in an oil drift model.

The new implementation includes realistic physical parameterizations of entrainment by breaking waves, realistic droplet size spectra, and a physical description of vertical mixing and resurfacing. Having a ready-to-use oil spill model that does not

require observation-based tuning is a requirement for operational modeling in real incidents. Only knowledge of the spilled oil type, its position, and environmental conditions from forecast models is required to perform operational simulations.

## 5   Ensemble simulations

In the following we present a series of oil spill transport simulations that serve as examples to illustrate how a hypothetical oil spill could develop during the first days after release. Focus is given on how the horizontal transport differs for various oil

types and weather conditions.

The Norwegian Sea off western Norway is selected, being a region that contains several oil fields that are currently exploited, in addition to shipping activity for industry, fishing, and commercial transport. Oil spill simulations for three different oil types are presented, of which two are oils that are exploited in this region and the third is a heavy bunker oil commonly used as fuel for shipping (Tab. 1).

To cover multiple possible outcomes of an oil spill during various weather and ocean current situations, ensembles of 72 overlapping periods of 10 days are simulated for each of the three oil types. In each ensemble, 10000 super-particles are released at the surface every second day and followed for 10 days using OpenOil.

Each ensemble member is initialized on different start dates that are uniformly spread throughout January to May 2011. Although the start dates are not random, the sampled weather and oceanic conditions are considered as random selection.

For the Lagrangian particle tracking model, each ensemble is effectively initialized with the same initial condition (e.g., the locations of the super-particles), but with different forcing data (i.e. weather and oceanic conditions).

The ensemble members are not designed as exemplary oil spill events with a distinct spill location but rather to provide widespread initial conditions that reflect many possible outcomes of arbitrary oil spills in the region. The initial spill location in the simulations are therefore set to a larger region with Gaussian spatial distribution around 3.5° E and 61° N with a standard

deviation of 20 km.

We attempt to answer how much of the oil in a spill in this region can be expected to become submerged into the water column and how much will remain or reappear at the surface. We also identify from the ensemble the main directions of transport for possible oil spills. The analysis yields an expected average oil budget for the first ten days after an oil spill, and we discuss how the oil budget differs for various oil types.

### 5.1   Ensemble results

To illustrate the behavior of the given oil types in various weather conditions, we first present six examples out of the 216 ensemble simulations. Fig. 2 shows three simulations of the same date and weather condition, but for the three different oil



types. For each simulation, a map with the initial positions and the final positions after 10 days are shown, along with the oil budget, wind and wave conditions, and oil density and viscosity. The oil budget illustrates how much oil is evaporated, stranded, submerged or at the surface.

The cases in Fig. 2 show that the three oil types result in quite different drift patterns, given the same environmental condi-
tions. The weather during this 10-day period is marked by a severe storm during the first days, followed by a period of light wind and a second short storm event. The light oil in case I(*VISUND*) is mostly submerged during the strong winds, but during calm conditions with low wind speed some of the oil resurfaces. The oil is again fully submerged when the wind picks up for the second time.

The medium oil in case II (*OSEBERG A*) gets fully submerged during the first days, but resurfaces quickly. Different from the
light oil, it is not much entrained during the second storm. The high viscosity of the emulsified oil prevents wave entrainment at this stage. The heavy bunker oil in case III *IFO-180LS* gets barely entrained even for wind speeds up to 15 m/s. Some oil is entrained during the first storm, but not during the second when the oil is aged.

The horizontal transport patterns reflect how much oil is at the surface and how much is submerged. The surface oil slicks of the medium and heavy oil advance far towards the northeast, as they are affected by winds and waves. The submerged oil (i.e.,
light oil and parts of the medium oil) are spread more homogeneously, and move more slowly with the ocean currents. Parts of the submerged oil also moves toward the northwest.

The examples in Fig. 3 illustrate the behavior of one oil type (*OSEBERG A*) during three different weather conditions. The three cases are marked by various degrees of wave entrainment during the first hours of the simulation. Case I in Fig. 3 shows a situation with almost no submersion during the first days. Even though the wind picks up later, no oil is submerged
as it is emulsified at later stages. The surface slick oil is transported northeast, and mostly organized in narrow bands that reflect regions of confluence in the oceanic flow. The oil in case II and III is submerged during the beginning of the simulation and resurfaces later, resulting in lower transport velocities towards northeast and more transport by ocean currents towards northwest. In general, the examples in Fig. 2 and 3 show that submerged oil tends to spread more evenly, as it is not constrained by converging flow fields at the surface level.

## 25  5.2  Composite oil spill budget

Figure 4 shows composite mass budgets for each of the three oil types in Table 1. These mass budgets show the sum over all ensemble members for the respective oil type, such that the effects of individual weather events are removed. The composite budgets depict the average time development of an oil spill in the study region, given typical weather situations from January to May.

Figure 4a shows that most of the heavy bunker oil *IFO-180LS* remains at the surface during the course of 10 days. Little oil is evaporated or submerged. A substantial amount of this oil at the surface is stranded after about 2-10 days following release.

The medium density oil (*OSEBERG A*, Fig. 4b) develops a mass budget with similar amounts of evaporated, stranded, submerged and surface oil. Stranding occurs later as for the bunker oil, and to lesser degree. More than 60% of oil is submerged during the first hours, but most re-surfaces during the following days.



For the light oil (*VISUND*, Fig. 4c) about 50% of the mass evaporates. Submersion during the first hours is high as for the medium oil, but with less re-surfacing such that most oil that is not evaporated remains in the water column.

## 5.3 Droplet size distributions

Although the droplet size spectra resulting from the stochastic wave-entrainment events are a direct function of oil properties
and environmental conditions (Eq. 12), the droplet size spectra of all submerged oil changes over time for two reasons: Firstly, oil properties change during the simulation, resulting in different spectra during new entrainment events. Secondly, large and small droplets rise at different speed to the surface. Resulting droplet size spectra, summed over all simulation ensembles, are shown in Fig. 5. The heavy bunker oil generally exhibits larger droplets then the medium and light oil. For all oil types, the droplet size spectra is shifted towards smaller droplets during the first 3 days. The spectra are also becoming narrower because
large droplets become relatively less abundant in the water column as they quickly resurface.

The gradual removal of large droplets from the water column is also evident from Fig. 6, which shows the vertical distribution of oil mass after one hour (panel a), 3 hours (panel b) and 24 hours (panel c). The blue and red distributions are for large and small droplets, respectively. Both classes are mixed down to equal degree after one hour, but the most of the large droplets have risen to the surface after 24 hours (panel c), while many of the small droplets remain dispersed. For deeper layers below
the mixed layer (here about 50 m), the amount of small particles steadily increases during the first 24 hours. This mechanism is also demonstrated by Moghimi et al. (2017, Fig. 1).

## 5.4 Horizontal transport

Composite concentration maps after 10 days of simulation are shown in Fig. 7, separated into oil types and surface vs. submerged oil . The composite concentrations are calculated as the sum of the ensemble members, hence removing the impact of
individual weather events.

The surface slick of all oil types (Fig. 7a,c,e) generally follow the Norwegian Coastal Current and North Atlantic Current towards the northeast. The general wind and wave pattern is also towards the northeast in this region. However, the surface slick of the heavy bunker oil moves the furthest, and the light oil shows the least expansion towards the northeast.

After 10 days, little of the heavy and medium density oil is submerged (Fig. 7b,d). A substantial amount of light oil (Fig. 7f)
is submerged after this period, which exhibits a strong branch towards the northwest.

An attempt to interpret the pathways of possible oil spills of western Norway is given as follows. Figure 7 shows that most of the surface slick follows the average currents towards the northeast along the coast of Norway. These are the North Atlantic Current and the Norwegian Coastal Current. The average wind and wave direction support movement in this direction, also causing intermediate stranding along the coast. Figure 4 therefore shows the most stranding for heavy oil, followed by the
medium oil.

The surface slick and the submerged oil constantly exchange mass with each other (section 4 and Jones et al. (2016)). This explains why the surface slick of the heavy bunker oil moves fastest towards the northeast; this oil type spends the most time at the surface (compare section 5.2 and 2.4) and is therefore more exposed to wind drag and Stokes drift. The surface slick of the



light oil is composed of particles that have spent more time in the water column, and hence is transported more slowly towards the northeast (Fig. 7c).

The submerged light and medium oil (Fig. 7d,f) moves partly towards the northeast with the average ocean currents. Another branch of the submerged oil is directed towards the northwest, which is most likely due to eddy transport because the average

current pattern does not support transport in this direction. Strong baroclinic eddies are a steady feature along the currents that lead towards the northeast, which can entrain particles towards the northwest (e.g., Hattermann et al., 2016). Some of the surface slick oil is also found towards the northwest. These surface slicks are likely composed of re-surfaced oil that has been transported northwesterly as submerged oil.

## 6 Discussion

Drift at the ocean surface generally differs from the drift just below the surface and from transport in deeper layers (Röhrs and Christensen, 2015). An oil drift model therefore depends on the accurate description of the mass exchange between surface slicks and submerged oil droplets. While the submersion of oil from the surface is mainly controlled by wave breaking and oil viscosity, resurfacing through buoyancy is controlled by the size of oil droplets and limited by oceanic turbulence (Sundby, 1983; Elliott, 1986; Tkalich and Chan, 2002). The examples in this study provide insight on how the vertical transport processes

affect the horizontal drift.

### 6.1 Model evaluation and performance

Two alternative parameterizations for the oil droplet size spectra have been considered in this study. Both are based on field- and laboratory experiments. In (Li et al., 2017b), a maximum droplet diameter based on Rayleigh-Taylor instability limit and the non-dimensional Weber and Ohnesorge numbers are used as parameters to express wave entrainment and the droplet size

spectra. The alternative parameterization (Johansen et al., 2015) is based on the Weber number and Reynolds number, and additionally requires knowledge on the oil film thickness. There is evidence that the oil film thickness has a strong impact on the droplet size spectra (Zeinstra-Helfrich et al., 2016). However, in the ensemble simulations for this study the formulation of Li et al. (2017b) is preferred because the amount of leaked oil is not specified in these experiments, hence oil film thickness is not a modeled variable. Using the Rayleigh-Taylor length scale instead of oil film thickness has the advantage that it is

generally easier to determine, as the film thickness is often difficult to model. In cases where the oil film thickness is well known or estimated, it may be advantageous to make use of the parameterization by Johansen et al. (2015).

The experiments is Sec. 4 are carried out with both droplet size spectra, and estimation of the oil film thickness was possible because the amount of spilled oil for this case is known. The droplet size spectra of Li et al. (2017b) provided a more robust model that is not as sensitive on fine-tuning of the unknown parameter, the emulsification rate. The droplet size spectra of

Johansen et al. (2015) was more sensitive to the emulsification rate and provided poorer match with observations without fine-tuning, but it also gave the most exact simulation when the emulsification rate was tuned to the observations. The oil film thickness was here estimated as 0.7 µm, assuming that all the spilled oil was spread uniformly within the slick area detected





by UAVSAR. However, as was observed from the ship, oil was not uniformly spread, but rather collected in patches. A general rule of thumb says that "90 % of the oil is located in 10 % of the area" (Hollinger and Mennella, 1973), which would implicate that the thickness in the thicker part is a factor of 100 thicker than in the thinner, larger areas. In fact, by increasing the thickness from 0.7 μm to 70 μm, the simulations using the Johansen et al. (2015) droplet spectrum compare better to observations (not

shown), about similar to the simulations using the Li et al. (2017b) spectrum for the same water content. Whereas we do not attempt to determine such a "patchiness-factor" here, we conclude that it makes an impact on the results, and that future work on its quantitative estimation is encouraged. Some work has already be done by Simecek-Beatty and Lehr (2017) on the redistribution of oil slicks due to Langmuir circulation cells.

How well the new parameterizations relate to different types of oil spills in the open ocean at variable environmental con-

ditions must be tested. One concern is the applicability to the larger scales of the real ocean, compared to the laboratories and wave tanks which have been used for model development and prior testing. Also, the interaction among various processes and mechanisms in oil spill transport may in principle yield different results than what is obtained in more controlled laboratory experiments.

The presented model provides a tool for integrated oil spill transport studies. In section 4 we have shown that the complete

oil drift model can simulate the time development of an observed oil spill where vertical exchange mechanisms play a major role, e.g,. wave entrainment of surface oil and resurfacing of submerged oil. A good match with observations is obtained from the model without tuning of model parameters. However, the agreement still improves by tuning of the emulsion water content, which is uncertain in the experiment.

The model formulation with an explicit formulation for wave entrainment, vertical mixing, and resurfacing also provides

a good description of the spreading of oil in windy conditions, as shown in Sec. 4 and discussed in Galt and Overstreet, R. (2011). For circumstances where spreading is not the cause of pressure-gradient forces, as, e.g., during the initial minutes after large oil spills, (Spaulding, 2017) conclude that the explicit formulation of wind-driven spreading is advantageous to otherwise common random walk approaches.

## 6.2   Vertical transport processes

The ensemble experiments reported in section 5 show that the submersion of oil from the surface into the water column greatly varies as function of oil type, weather situation and weather history. Mechanisms such as wave entrainment, emulsification and oceanic turbulence play a key role. With an exception for the heavy oil type, much of the oil becomes submerged during the first hours after the oil release, provided there is sufficient wind and waves. There is a continuous mass exchange between the surface slick and submerged oil, and much of the bulk mass returns to the surface after 2-5 days, when the oil is more

emulsified (Fig. 4). The aged (i.e. emulsified) oil forms larger droplets (Fig. 5) and thus resurfaces faster. Most of the larger droplets return to the surface within one day (Fig. 6).

Particular cases of the ensemble experiments also show that the weather during the first days of an oil spill affects the mass budget after 10 days (Fig. 3). If strong winds prevail during the first hours of an oil spill, large quantities of oil are submerged before evaporation occurs, prohibiting emulsification. Such oil is likely to stay submerged for longer time. Conversely, if calm




conditions are present during the beginning of an oil spill, evaporation of light components enables stronger emulsification and less submersion even if the wind increases during subsequent days.

The experiments show that vertical mixing is another key process in oil spill transport. Oil that is submerged by wave entrainment is mixed vertically due to oceanic turbulence. Large droplets typically resurface within hours or days (Fig. 6, blue curve) and the smallest droplets experience insufficient buoyancy to counteract oceanic turbulence (Fig. 6, red curve). In fact, large amounts of droplets on the smaller range of droplet diameters (70-250 μm) have been observed during dives using *Holograms* (Davis and Loomis, 2014). In a steady state as approximated by Eq. 11, the ratio of terminal velocity over eddy diffusivity regulates how deep the droplets are dispersed into the water column. This ratio will also set the time scale for resurfacing of oil. The terminal velocity depends mainly on droplet size, and the eddy diffusivity is largely determined by wind history, wave energy dissipation, and ocean stratification. It is hence advantageous to provide particle transport models with more realistic eddy diffusivities from ocean circulation models.

Wave entrainment is the only mechanism that submerges oil from the surface slick into the water column. On the open ocean, wave entrainment occurs through whitecapping events. The whitecap coverage is therefore a key parameter in wave entrainment parameterizations. Unfortunately, all of today's formulations for wave entrainment of oil use wind speed and significant wave height as a proxy for whitecap coverage and wave entrainment. In (Li et al., 2017c), a threshold of 5m/s wind speed sets the minimum wind speed activating whitecapping. Other parameterizations for whitecap coverage use lower wind speeds (Callaghan et al., 2008). In general, whitecap coverage and wave entrainment should be parameterized using wave energy dissipation. Wind speed and wave height do not provide sufficient knowledge of the wave energy spectrum to account for ocean swell dissipation, fetch-limited seas, or wave growth (Hasselmann, 1974; Melville, 1996). Wave energy dissipation is the mechanism that triggers whitecap events, and it is a prognostic parameter of wave forecast models (Komen et al., 1994) that could be used in oil spill modeling. However, no such parameterizations exist as of today, and replacing whitecap coverage in existing parameterizations for wave entrainment of oil is not straightforward. Co-located observations of whitecapping and wave entrainment of oil in the open ocean are needed to revise the existing methods. This weakness in wave entrainment parameterizations should be addressed in future work.

## 6.3 Horizontal transport

Details of the entrainment rate parameterization have previously been shown to affect the results for horizontal oil transport in idealized simulations (Li et al., 2017c). Our study shows considerable differences in entrainment rate between heavy (high viscosity) and light (low viscosity) oils when using the Li et al. (2017c) entrainment rate. A larger fraction of light oil is entrained compared to heavy oil. This affects the horizontal drift: fewer of the oil elements drift with the surface winds and Stokes drift, as a larger fraction is entrained and drifts with only the ocean currents.

The ensemble simulations in section 5 show that surface slicks and submerged oil often are transported towards different directions. Only the surface slick is exposed to wind and only near-surface particles are exposed to the Stokes drift. Therefore, the average transport pattern of oil depends on its potential for submersion. Oil that tends to remain at the surface, as, e.g., heavy bunker oil and emulsified medium density oil, is much more exposed to the Stokes drift and wind drag.





While ocean currents can transport oil to the near-shore, only wind and Stokes drift can move the oil onto the shore and cause stranding. Röhrs et al. (2014) show that the Stokes drift has a tendency to move buoyant particles towards the shore, and that submerged particles are sheltered from this shore-ward transport. To realistically simulate stranding, an entire chain of physical processes needs to be described: wave entrainment, vertical mixing, formation of oil droplets, resurfacing and the respective

advection by wind, wave and currents. Stranding then occurs when shore-ward wind occurs while surface oil is present in the near-shore, leading to very intermittent occurrence of severe stranding events (Nordam et al., 2016).

The ensemble simulations in this study show that droplet size spectra for the submerged oil change over the course of 10 days (Fig. 5). Laboratory experiments have also shown that continuous wave breaking can lead to a bi-modal droplet size distribution (Li et al. (2009, 2017b)), and that the number of large droplets decays with time. This behaviour is captured in

OpenOil, partly because the droplet size spectra depend on the emulsification rate, but also because the vertical mixing scheme lets large droplets resurface faster. (Li et al., 2017a) also use the vertical advection-diffusion equation to show that turbulent mixing and buoyancy leads to changes in the droplet size spectrum over time towards smaller particles in the water column, which corresponds to their experimental data for oil with low dispersant concentration.

Resurfacing of submerged oil is seen in all simulations performed for this study. Small and medium size droplets first

resurface when oceanic turbulence decays, which may take many days. This has been observed during the Golden Trader oil spill in the Skagerak Sea in September 2011 (Broström et al., 2014). Therefore, small droplets should not be disregarded in oil spill simulations. Instead of using a cutoff droplet size to set small particles as dispersed and large particles to remain at the surface, as is common in some oil drift model descriptions (e.g., Li et al., 2017c; Moghimi et al., 2017), we use vertical diffusion and buoyancy to resolve the resurfacing of particles explicitly. Environmental conditions then determine when particles of a

given size re-surface in a physically realistic way.

Emulsification greatly affects the vertical exchange between surface slick and submerged oil, and thereby also the horizontal transport. In the short term, the emulsified surface slicks are more prone to wind- and wave drag, as shown by the sensitivity tests in Sec. 4 (Fig. 1). In this test, knowing the degree of emulsification would help to simulate the observed oil slick. In the long term, the emulsified oil that is mostly near the surface also has an increased potential for stranding. However, weathering

of oil remains difficult to describe. Often, oil drift models are initialized without a-priori knowledge of the emulsification rate. In addition, the weathering module used in this study describes emulsification as a function of time that has passed since the release of oil. However, for a typical real case where an oil drift simulation is initialized from an observed surface oil spill, the observed oil will often be already aged (i.e. evaporated and emulsified), and thus an oil drift simulation should take this into account. Ideally, wave dissipation energy from a wave forecast model should be used to describe the process of emulsification,

similar to the issue of the wave entrainment rate discussed above. Future studies should address this using field experiments that include observations of wave breaking.

The physical mechanism behind the additional wind drag for the oil slick that is commonly used in oil spill modeling remains unclear and is a subject of investigation. Laxague et al. (2018) argue that the wind drag factor mostly compensates for the lack of vertical shear in the upper few centimeters of ocean models, which arises because the oil slick dampens capillary wind

waves. Additional motion in the downwind direction may also be the cause of wave-current interaction under the oil slick.



Due to momentum conservation, this dampening accelerates a current below the oil slick in the wave propagation direction (Christensen and Terrile, 2009).

## 7    Conclusions

A comprehensive oil drift model for the transport and weathering of oil spills in the open ocean, OpenOil, has been developed
and compared with the time development of an observed oil slick. The model is based on the Lagrangian particle tracking model OpenDrift (Dagestad et al., 2018), the ADIOS oil database and weathering library from NOAA, and a number of recent parameterizations which describe the mass exchange between the surface slick and submerged oil. Functionality from OpenDrift is inherited by OpenOil, including particle advection, time-stepping, and reading and interpolation of environmental input data from ocean, wave, and atmospheric models. OpenOil is available to the community as open source software  and is
used for operational oil spill modeling at the Norwegian Meteorological Institute.

The vertical exchange of oil spills between the surface and the water column are controlled by a chain of processes from wave entrainment to the resurfacing of oil droplets. Therein, the emulsification of oil and the resulting shape of the droplet size spectra play a major role to determine the distribution of oil between the surface and the water column. Using state-of-the-art parameterizations for wave entrainment and droplet size spectra (Johansen et al., 2015; Li et al., 2017c, b), OpenOil is able to
simulate the spreading and transport of an oil slick that has been observed for 7.5 hours after release.

As a result of the vertical processes, the horizontal transport of oil spills depend on oil type, weather condition, and oceanic turbulence. Heavy oil that tends to remain at the surface is subject to rapid transport by winds and waves and more prone to stranding. Lighter oil types become largely submerged if the weather conditions enable wave entrainment. While the submerged oil is transported at lower speeds with the ocean current at depth, oil spills that are partly submerged typically follow wind
and wave directions to a smaller degree because most particles repeatedly re-surface and get re-entrained. Resurfacing of submerged oil after a couple of days is common for both heavy and light oil types. These results show the importance of 3-dimensional transport modeling including weathering processes and the specific oil characteristics for realistic simulation of the drift and evolution of oil spills, and the sensitivity of long-term transport on the initial weather conditions at the time of release.

*Code and data availability.*    The oil spill model used in this study is openly available on www.github.com/opendrift. Ocean and atmospheric model data is available on thredds.met.no. Data from field experiments are available on request to the corresponding author.

*Author contributions.*    KFD developed the model OpenOil with contributions from JR, HA and TN. KFD carried out the model simulations in section 4. CB, CEJ, and KFD designed and carried out the field experiments in section 4. JR designed and carried out the numerical

---

[2] http://www.github.com/opendrift

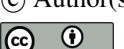



experiments in section 5 with contributions from KFD, TN and JS. TN and JS provided background knowledge on oil spill modeling. JR prepared the manuscript with contributions from all authors.

*Competing interests.* The are no known competing interests associated with the investigations and results presented in this paper.

*Acknowledgements.* This work has been funded by the Research Council of Norway through grants 244626 (RETROSPECT) and 237906
5  (CIRFA), and in part by a grant from The Gulf of Mexico Research Initiative (award "Influence of river induced fronts on hydrocarbon transport", GOMA 23160700). This research was carried out in part at the Jet Propulsion Laboratory, California Institute of Technology, under contract with the National Aeronautics and Space Administration. The UAVSAR data are courtesy of NASA/JPL-Caltech and are available through http://uavsar.jpl.nasa.gov or the Alaska Satellite Facility (www.asf.alaska.edu).





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





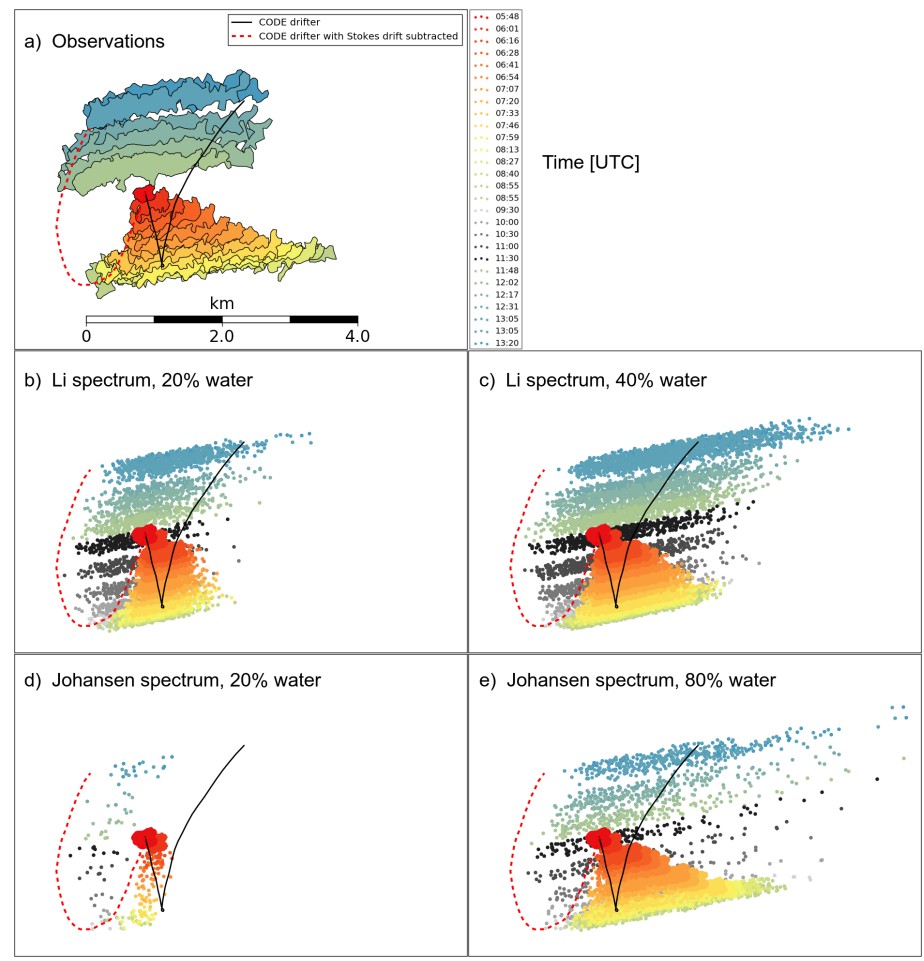

**Figure 1.** a) Oil slick contours observed from UAVSAR. b) Modeled oil slick contours using droplet size spectra from Li et al. (2017b), with a water content of 20%. c) same as b), but with a water content of 40%. d) Modeled oil slick contours using droplet size spectra from Johansen et al. (2015) with a water content of 20%. e) same as d), but with a water content of 80%. Only modeled oil elements at the ocean surface are shown here, for direct comparison with the observations. The black line denotes the trajectory of a CODE drifter. The dashed red line is a progressive vector diagram of the CODE drifter velocities with the Stokes Drift from the operational wave model subtracted. This velocity is used for the oil drift simulations.







**Figure 2.** Three oil drift simulations with different oil types, using the same start date (22 Feb 2011). Panels Ia-d shows a simulation with a light oil (*VISUND*), panels IIa-d shows a simulation with medium oil (*OSEBERG A*), and panels IIIa-d shows a simulation with heavy bunker oil (*IFO-180LS*). For each simulation, panel a) shows the oil mass budget, panel b) shows the mean and standard deviation of wind and wave conditions over all particles, panel c) shows the mean and standard deviation of the oil density and viscosity and panel d) shows a map with the initial particle positions (green), positions after 10 days of transport (blue), stranded particles (red), and particle trajectories (gray lines).







**Figure 3.** Three oil drift simulations with different start dates, using the same oil type *OSEBERG A*. The start dates between each simulation are 2 days apart and each simulation lasts for 10 days. The panels are organized as in Fig. 2.





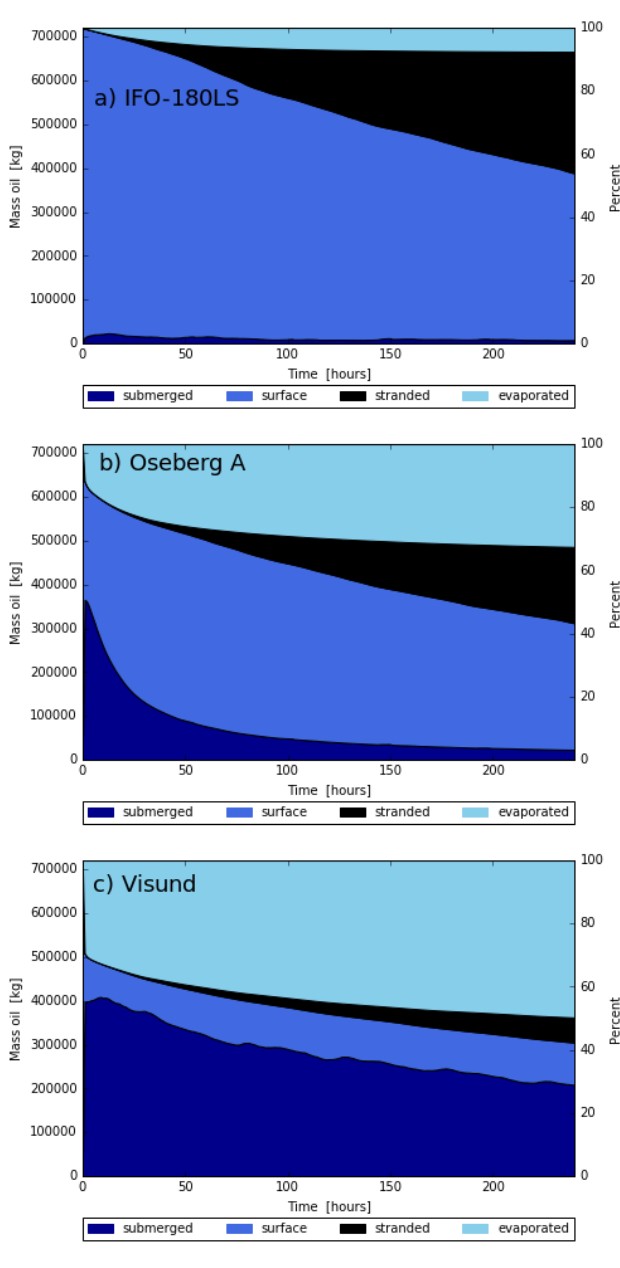

**Figure 4.** Composite mass budgets for 10-day simulations of 3 different oil types, based on 72 ensembles for each oil type.





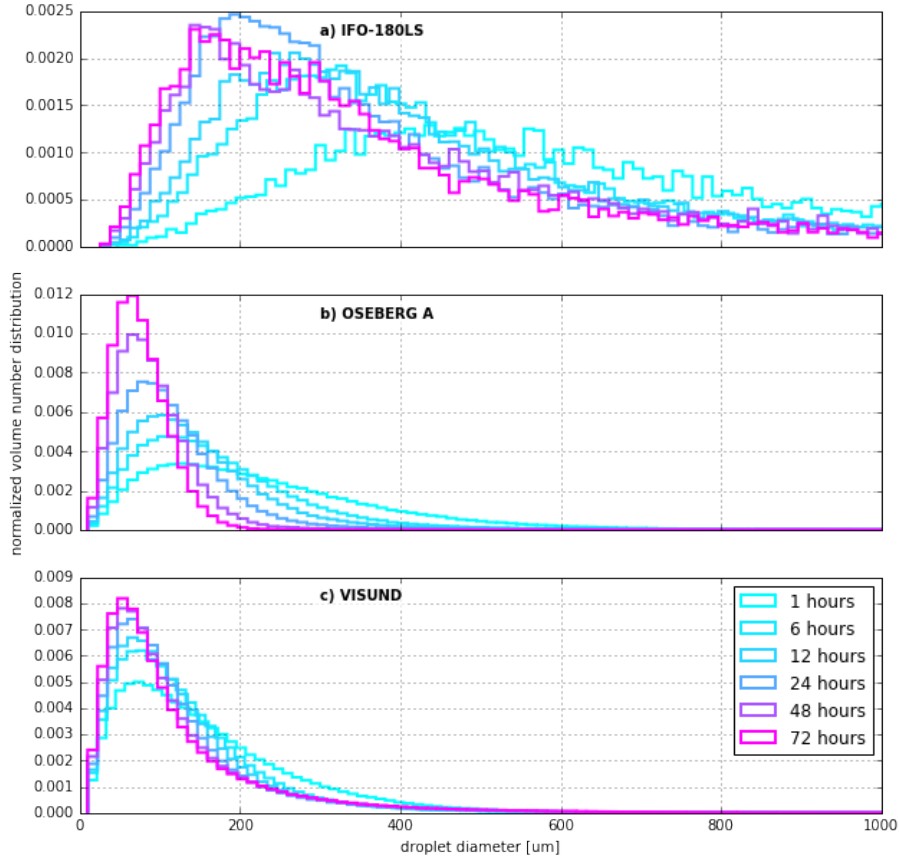

**Figure 5.** Droplet size mass distributions for 3 different oil types, based on 72 ensemble simulations for each oil type. The colors in each plot show the temporal evolution of the droplet size distribution.

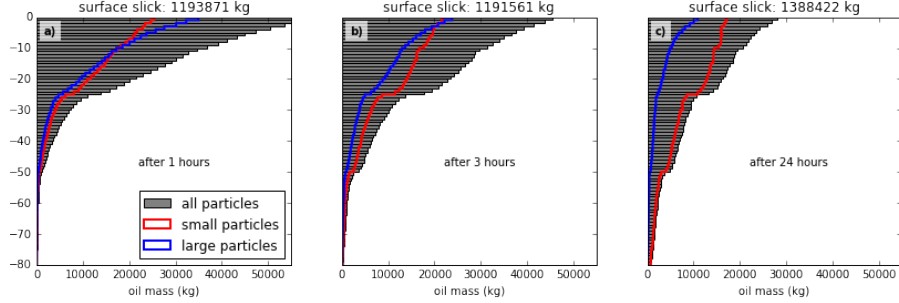

**Figure 6.** Vertical mass distribution of submerged oil droplets for all particles (shaded grey), small droplets (red) and large droplets (blue). The overall medium droplet diameter from these simulations (160 $\mu m$) is used as threshold between small and large droplets. a) shows the vertical distribution after one hour, b) 3 hours, and c) 24 hours. All 3 oil types are included in these distributions.





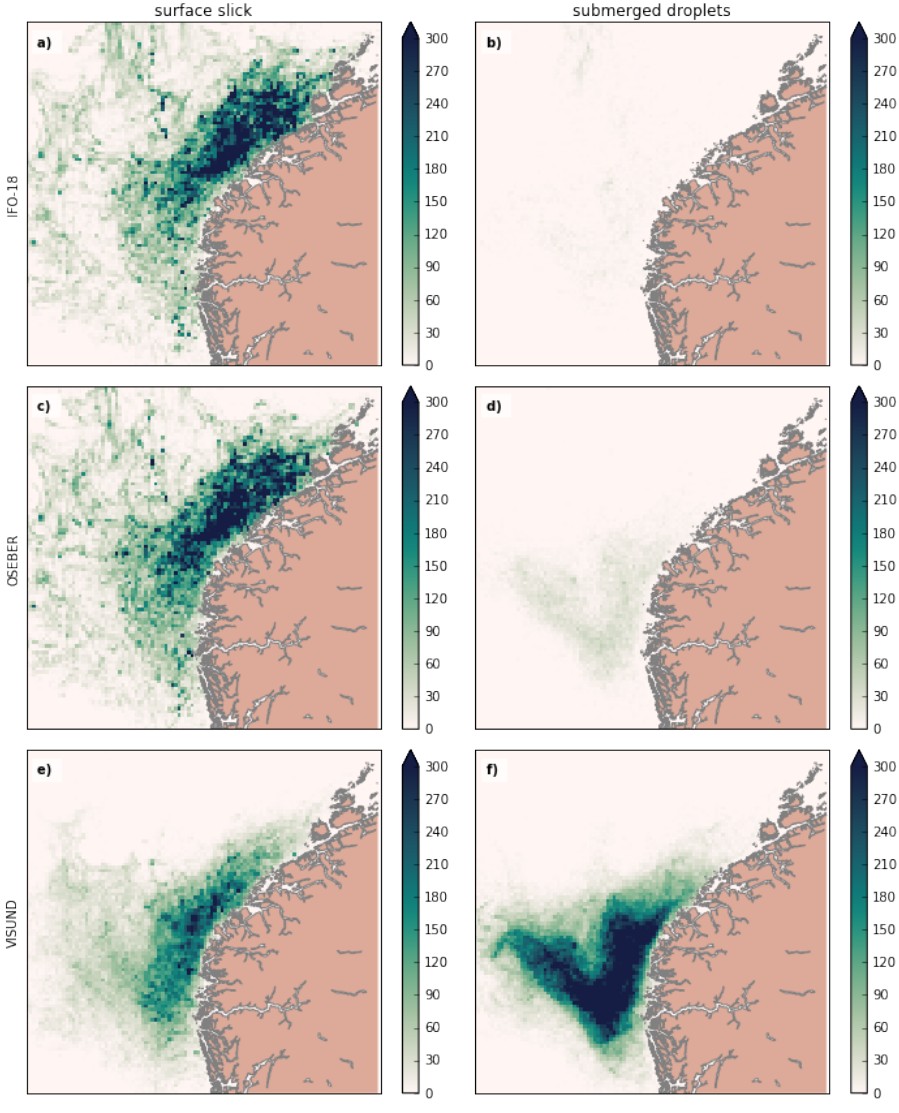

**Figure 7.** Horizontal concentration maps for stochastic oil spills simulations after 10 days, summed over 72 simulations for each oil type. Submerged oil (right column) and surface oil (left column) are displayed separately.





**Table 1.** Oil types used for the oil spill transport simulation of spills off the coast of western Norway. Visund and Oseberg A represent light and medium density oil types that are exploited in the study region. The IFO-180LS is included as example of heavy oil (bunker oil), used as fuel for shipping.

|  | Oil type | density [$kg/m^3$] | viscosity [$mPa \cdot s$] | reference |
|---|---|---|---|---|
| light | Visund | 791 | 2 | Sorheim (2009) |
| medium | Oseberg A | 902 | 53 | Strom (2013) |
| heavy | IFO-180LS | 973 | 7426 | Sorheim (2014) |