# Peer review of "The effect of vertical mixing on the horizontal drift of oil spills"

_Ocean Science, 2018_

## Referee Comment (RC1) · N. Laxague (Referee) · 9 Oct 2018

True to the title, the authors present an investigation of the horizontal drift of oil using numerical simulations, focusing on the dependence of transport on vertical mixing processes. The first part of the manuscript makes use of the dataset described in Jones et al. [2016] in order to qualitatively validate the OpenDrift modeling framework given differing droplet size spectra and emulsion characteristics. The rest of the analysis rests in the model's description of three different simulated oil spills.

The topic is appropriate for Ocean Science Discussions. Its principal value seems to be in its building on the Jones et al. [2016] results and generalization toward real-world oil spill transport. I have a few comments on the way that the validation phase of the

study is presented, but ultimately feel that the manuscript is sound and represents a useful advance in the field.

1. It would be useful to move beyond the qualitative comparison of observed and modeled slick extent. How do we know that one droplet size spectrum (i.e., Johansen or Li) matches observations better than the other? It is stated that they were introduced in order to eliminate the need for model tuning. But isn't that what ends up happening as the emulsion characteristics in the model are adjusted to match SAR observations of slick extent?

2. It is stated in Jones et al. [2016] and in the present manuscript that the surface expression of oil is largely a creature of subsurface emulsified oil which is generally not transported by Stokes drift. Can you comment on the relative importance of this mode of transport vs. surface oil being moved by wind/Stokes drift? It seems that overall surface slick transport is a combination of the two. This leads directly to my final comment:

3. It is mentioned that SAR (P12, L30) does not show the whole downwind extent of the slick (as determined from a visual observer). This provides some context to the shortening of the slick line seen by SAR in between flights as shown in Figure 1. Does this merit any amount of reconsideration of the UAVSAR's status as the principal observational reference point? Do you think that the underestimation of downwind slick extent is causing the subsurface oil component to be overstated as a mode of surface slick transport?

---

## Referee Comment (RC2) · Anonymous Referee #2 · 16 Oct 2018

The Authors present an interesting study of modeling vertical and horizontal transport of oil spills in the Norwegian coast using a open source oil drift model. The study includes improvement of modeling system, validation of the new model results and model simulation with different types of oils and oil drift in different weather, wave and current conditions. The model developments and results are thoroughly presented. However, at places the text is quite descriptive, and one would expect more in depth analysis of the results. For example, in section 4.3 more discussion about the reasons behind the different behavior of the two approaches would be interesting. Also some discussion on how the need for tuning of the water content would affect the use of the model system in operational forecasting. In section 5 more description and discussion about the seasonal and inter-annual variation in the weather conditions in the study area is

needed to convince the reader, that the selected period actually represents the typical weather conditions in the area. Otherwise, the selected period should be presented as case study and give an indication, why the Jan-May 2011 period was selected. In section 5 and in Conclusions it would also be good, if the Authors would more clearly state, which part of the described behavior of different oil types was expected based on previous studies and what is actually knew knowledge gained by this study. I also suggest that the spelling and language is checked before accepting the manuscript.

Some specific comments:

Section 1:

In some places the text seems more a list of references (for example page 2 lines 9-22) than description of background and earlier studies. I suggest that some rewriting is done for this section.

Section 3:

What are the thicknesses of vertical layers in the Norshelf reanalysis?

What is the horizontal resolution of the ERA-interim wind and wave fields.

Subsection 3.3 could be merged to 3.2

Section 4:

Fig 1. It is difficult to see the red dashed line (CODE drifter with Stokes drift subtracted) printed above the read color representing the particles in the beginning of the measurement period.

Page 11, lines 4-5, the information about the gray colors should be in the caption of Fig. 1

Page 11, line 19: What is the resolution of the WAM Stokes drift fields?

Page 11, lines 21-23: Which parameters were measured from the ship? And which

parameters were estimated? What was the bias between measured and modeled values? And how the accuracy of the models affect the results presented in section 4.3?

Page 11, line 28: Is the beginning of this line, reference to Sec 2.1?, misplaced?

Page 12: How sensitive are these two approaches to the accuracy of the forcing wind and wave field?

Page 12: The accuracy of the drift simulation seems quite sensitive to the emulsion water content. How do you see, that this affects the model behavior and use in operational situations, where the tuning cannot be made?

Section 5:

It was said somewhere earlier in the text that in section 5 you will use parameterisation of Li et al. Maybe this information could be repeated in the beginning of this section.

Is there a specific reason for choosing year 2011 for the simulations? How much seasonal and year-to-year variation is there in the weather and ocean current conditions in this area?

Page:14, lines 28-29: If you are only using forcing data from one year, how have you defined that they represent typical weather situations from January to May.

Section 6:

Page 18, line 3→ How sensitive is your analyze to the accuracy of the forcing models in describing the conditions, especially the vertical mixing?

Section 7:

It would be good to emphasize in the conclusions what is general knowledge in the behavior of different oil types and what are your findings.

---

## Author Comment (AC1) · 2 Nov 2018

We would like to thank for a constructive review. Addressing the first point raised by the reviewer, we agree that a quantitative comparison of the model simulation with the measured oil spill contours would be better than a purely visual inspection of the results. We are therefore working on a way to subjectively evaluate the model simulations with the various droplet size spectra and emulsion rates. This additional analysis will be presented in the revised manuscript.

The reviewer also addresses a weakness in section 4 that we end up tuning the model by selecting the water content in the oil emulsion that provides the best match with observations – and we do not see a way to overcome this issue in the present study.

The need for tuning the water content arises because the oil in the experiment was pre-emulsified before spilling the oil, and we do not know how the oil then continues to emulsify in the open ocean. The used algorithm for emulsification does not account for this situation. Nevertheless, using realistic droplet size spectra has greatly reduced the need for model tuning compared to the previous model implementation used in Jones et al. (2016). We will clarify this in the revised manuscript and add to the discussion.

Considering point two, we should clarify that Stokes drift transport is relevant for both surface and subsurface oil, but for the subsurface oil the effect is much smaller and dependent on the depth. Below the Stokes layer, it should be negligible. This leads to the explanation that parts of the surface slick, which consists of resurfaced oil, lags behind the part of the surface slick that has experienced continuous transport by wind and waves.

While the UAVSAR underestimates the total extension of the downward slick, the parts that are not seen are expected to be very thin, such that they do not provide sufficient alteration of wind ripples to be detected by the image processing. For the total mass budget of the spill, this part may be less relevant than the bulk of the oil in the trailing part of the oil slick. However, this may be different for light and heavy oil which is submerged to a different degree. We will include this consideration in our discussion of results.

OSD

---

## Author Comment (AC2) · 2 Nov 2018

We would like to thank the reviewer for a thorough feedback on our manuscript. We will be happy to provide a more detailed discussion on why the two droplet size spectra give different results. While it is difficult to say something conclusive about the two parameterizations, we think that a key practical difference is that the Li et al. (2017) spectrum gives a more robust model system that is less sensitive to uncertainties in the emulsification rate and slick thickness (the latter is not a parameter in this parameterization). The Johansen et al. (2015) spectrum, on the other hand has a more explicit treatment of the oil slick properties. This may provide superior results if slick thickness and oil properties are well known, but is also more prone to errors when uncertainties are large, judging from our experience with the two methods (see e.g. Fig. 1).

[Figure]

We agree that the need for tuning the water content in the oil emulsion for the experiments in sec. 4 should receive more discussion in the paper. For operational purposes, our problem that we did not know how the pre-emulsified oil would behave after the spill, may not be a limitation if the initial oil spill is not emulsified from before. However, the same problem will arise when operational oil spill models are initialized from observed oil spills without knowledge of the time and location of the initial spill. In such cases, using the (Li et al. 2017) droplet size spectra can reduce uncertainties compared to other methods. In addition, we foresee a revision of the emulsification algorithm in future studies.

Considering section 5, it correct that we have not addressed inter-seasonal and inter-annual variations in the weather situations of our study region. For our study, an important requirement is that we cover a large range of different weather situations with regard to wind and wave forcing. The winter and spring season in the Norwegian Sea is charaterized by both stormy situations of passing lows (as during autumn) and calm weather with stable high pressures (as during summer). The simulations during January to May (one simulation starting every second day) did sample a large number of storms and calm situations, such that our set of experiments can be treated as Monte-Carlo simulations with random-like forcing. This allows us to map out most of the different possibilities for oil spill transport in the region, but we do not provide conclusions on inter-annual and seasonal variability of expected oil spill transport. This will be clarified in the revised paper.

A revised manuscript will include the discussion points above and we will also address all of the specific comments raised by the reviewer, particularly the suggestions on improving the style of the text in parts of the introduction.

---

## Author Response (AR1)

**Response to reviewer comments and changes to the revised manuscript:**

We would like to thank both reviewers for their constructive feedback. Answers to all comments and questions follow below:

**- Quantitative comparison of model simulations in Sec. 4:**

We now provide a quantitative comparison of the model simulations in Sec. 4, using the mean position and standard deviation of particle positions, compared to the observed slick in a new figure. This supports the identification of which of the simulations is considered a best fit with observations. Sec. 4 includes:

*"A quantitative comparison between the four simulations is presented in Fig. 2, where the difference in the mean position between the observed and modeled oil slick is evaluated in addition to the standard deviation of particles positions. While the mean position provides a measure for the overall horizontal transport of the oil slick, the standard deviation of particle positions provides a measure for spreading of the oil slick.*
*This quantitative evaluation of the model simulations allows for conclusions that are on par with the subjective interpretation of oil slick development from Fig. 1, i.e. the simulations with adjusted water content performs best with regard to spreading. The simulation using a droplets size spectrum from (Johansen et al., 2015) has has poorest performance with the non-adjusted water content (20%), and very good agreement with adjusted water content (80%). The simulations with the droplet size spectrum from (Li et al., 2017b) are less sensitive to such tuning, and provide very good fit with observations with regard to both mean positions and spreading, particularly for the first few hours of the experiment."*

**- Model tuning, pre-emulsification and consequences for operational models:**

The need for tuning the water content arises because the oil in the experiment was pre-emulsified before spilling the oil, and we do not know how the oil then continues to emulsify in the open ocean. The used algorithm for emulsification does not account for this situation. Nevertheless, using realistic droplet size spectra has greatly reduced the need for model tuning compared to the previous model implementation used in Jones et al. (2016), which largely depended on tuning the (constant) droplet size directly, as well as the entrainment length scale. For operational modeling, we now give a clear recommendation to use the Li et al. 2017 droplet size spectra, because it is less sensitive to fine-tuning of water content. In practise, this is important in operational modeling when the exact time of the spill is unknown, or if the oil spill model is initialised from observed oil slicks. We emphasize these points in par. 5 of section 4.3

*"The simulations with modified degree of emulsification, parametrized by the water content, provide an excellent fit to the observed oil slick contours of Fig. 1a, for both the first hours and the subsequent hours. Even if no modification of the water content is done, the model provides reasonable results when using the droplet spectra parameterization of Li et al. (2017b).*

*...*

*The new implementation of OpenOil is generally able to reproduce the oil spill transport observed during the oil-on-water experiment in Jones et al. (2016). While the previous model formulation was largely dependent on tuning of model parameters to the observations, the new model implementation can reproduce the oil slick transport qualitatively without a-priori knowledge. However, the results still benefit from fine-tuning of the water content, illustrating that the oil weathering and emulsification is a very sensitive component in an oil drift model."*

and furthermore comment on consequences for operational modeling in the end of Sec. 4.3:

*"A difficulty in operational oil spill modeling remains when the exact time of an oil spill is unknown, or if the oil spill model is  initialized from observed oil slicks. In these cases, the progress of oil emulsification may be unknown and need to be estimated. To reduce the model sensitivity that emerge from uncertainties in emulsification rates, one should therefore use the droplet size parameterization of Li et al. (2017b) in operational models, which is less sensitive to the exact water content."*

**- What is the differences between the two parameterizations?**

The major difference is that the Johansen et al 2015 parameterization explicitly uses the slick thickness, but it is also much more sensitive to emulsification. The parameterization of Li et al. 2017 does not use slick thickness and and is less sensitive to emulsification rate, hence it is more robust when these variables are unknown. This is now discussed in section 4.3 and 6.1, and briefly mentioned in section 7.

**- Stokes drift contribution to subsurface part of the oil spill (Sec. 4):**

This is clarified in section 4.3, second paragraph, and we now explicitly state that both parts are affected by Stokes drift but the surface Stokes drift is much stronger compared to the sub-surface part.

*"... Both parts are affected by the Stokes drift, but the surface Stokes drift is much stronger and decays rapidly with depth. While the western tip of the oil slick is a result of the ocean current and weak Stokes drift, the elongation of the surface slick towards the east is a result of wind and surface Stokes drift."*

**- UAVSAR underestimating the downwind part of the surface slick:**

The new figure (2b) also shows that the shortening of the observed slick line is associated with a reduction in oil slick spreading. Since spreading should be a continuously increasing process, we conclude that parts of the oil slick are not observed by UAVSAR during the later time steps. We have added this consideration in Sec. 4.3:

*"For the comparisons it is also important to note that very thin oil slicks do not provide sufficient alteration of wind ripples to be visible in SAR. In fact, it has been reported during the measurement campaign that the SAR observations did not show the entire oil slick that was visible from the nearby ship (Jones et al., 2016). This observation is supported by Fig 2b, showing a decrease in spreading of the UAVSAR oil slick during later times. As spreading should generally increase, we presume that the UAVSAR oil slick lacks parts of the downwind trail that is very thin. The UAVSAR images should therefore be interpreted as a measure for the majority of mass in the surface slick rather than the total area extent."*

**- interseasonal and interannual variation, Sec. 5:**

We agree that these experiments should be considered as case study. As such, the selected period provides for a wide range in possible current and and weather conditions. The relevant parts in Sec. 5 now read:

*In the following we present a series of oil spill transport simulations that serve as case study to illustrate how a hypothetical oil spill could develop in various weather conditions during the first days after release.*
*...*
*Each ensemble member is initialized on different start dates that are uniformly spread throughout January to May 2011. Although the start dates are not random, the sampled weather and oceanic conditions are considered as random selection to realize Monte-Carlo simulations with random-like forcing data. For the Lagrangian particle tracking model, each ensemble is effectively initialized with the same initial condition (e.g., the locations of the super-particles), but with different forcing data (i.e. weather and oceanic conditions). The winter and spring season provides for a wide range in circulation patterns with both calm high pressure weather situations and stormy periods with passing low pressure systems. Herein, we do not assess the inter-seasonal and inter-annual variability in oil spill transport for the study region.*

and if sec 5.2:
*"...The surface currents and wind patterns that are covered during this period cover a large range of possible situations for this regions."*

Replies to specific comments:

**>> In some places the text seems more a list of references (for example page 2 lines 9-22) than description of background and earlier studies. I suggest that some rewriting**

**is done for this section.**

We have partly re-written these parts of the introduction, and partly deleted parts that are repetitive with other parts of the paper. These paragraphs now focus on why oil droplet size spectra and vertical mixing might be important for the horizontal transport. The order of the paragraphs has also changed a bit.

**>> What are the thicknesses of vertical layers in the Norshelf reanalysis? What is the horizontal resolution of the ERA-interim wind and wave fields.**

We have corrected that ECMWF operational archives are used as forcing, i.e. atmospheric forcing is on 0.1 deg resolution (approx. 9 km). We have added this information in Sec. 3.1.

**>> Subsection 3.3 could be merged to 3.2**

We have merged these sections.

**>> Fig 1. It is difficult to see the red dashed line (CODE drifter with Stokes drift subtracted)**
**printed above the read color representing the particles in the beginning of the measurement period.**

We have modified the figure to emphasize the CODE drifter line.

**>> Page 11, lines 4-5, the information about the gray colors should be in the caption of Fig. 1**

Done.

**>> Page 11, line 19: What is the resolution of the WAM Stokes drift fields?**

This is now given in section 3.1, i.e. WAM fields are taken from the ECMWF operational archive at 0.125 deg resolution (approx. 14km).

**>> Page 11, lines 21-23: Which parameters were measured from the ship? And which parameters were estimated? What was the bias between measured and modeled values? And how the accuracy of the models affect the results presented in section 4.3?**

Wind was observed at the ship's mast and wave height were only estimated. Section 4.2 now includes more details on the weather observations:

*"The deviation between forecasted wind and observed wind from the ship was below 14% in magnitude, the difference in direction below 10 ◦ , and wave forecasts matched the estimated wave height during the experiment (details in Jones et al. (2016))."*

We see no indication that the quality of wind and wave forecasts affect the results for this sections. The errors associated with the oil emulsification and exact magnitude of the wind drift factor dominate the model errors in this case.

**>> Page 11, line 28: Is the beginning of this line, reference to Sec 2.1?, misplaced?**

Yes. We have resolved this error.

**>> Page 12: How sensitive are these two approaches to the accuracy of the forcing wind**
**and wave field?**

We do not see a difference for the two droplet size spectra with regard to sensitivity on wind and wave forecast errors. However, using the Johansen et al. droplet size spectra, the surface fraction of the oil spill is more sensitive to water content, therefore also to wind speed in general. Uncertainties in emulsification rate dominate the errors of the oil spill model.

**Page 12: The accuracy of the drift simulation seems quite sensitive to the emulsion water content. How do you see, that this affects the model behavior and use in operational situations, where the tuning cannot be made?**

see above.  A remark is added in sec. 4.3:

*"A difficulty in operational oil spill modeling remains when the exact time of an oil spill is unknown, or if the oil spill model is  initialized from observed oil slicks. In these cases, the progress of oil emulsification may be unknown and need to be estimated. To reduce the model sensitivity that emerge from uncertainties in emulsification rates, one should therefore use the droplet size parameterization of Li et al. (2017b) in operational models, which is less sensitive to the exact water content."*

and in the conclusion:

*"The performance of two*
*droplet size spectra is compared, finding that Li et al. (2017b) provide a more robust model for operational modeling, while the model of Johansen et al. (2015) can perform better in controlled experiments where oil film thickness and emulsification rate are known."*

**>> It was said somewhere earlier in the text that in section 5 you will use parameterisation**
**of Li et al. Maybe this information could be repeated in the beginning of this section.**

Done.

**>> Is there a specific reason for choosing year 2011 for the simulations? How much seasonal and year-to-year variation is there in the weather and ocean current conditions
in this area? If you are only using forcing data from one year, how have you
defined that they represent typical weather situations from January to May.**

The year 2011 is selected for practical reasons because the used ocean reanalysis had been available starting from 2011. The winter and spring is selected because the time period contains a wide range of forcing conditions, i.e. calm situations and a few severe storms. While there is year-to-year and interseasonal variation in the hydrography (temperature and salinity) in the region, ocean current statistics do not vary from year to year (at least not in the model). We argue that statistics of surface currents and wind patterns cover the majority of the possible situations for the region. We have added this view at the beginning of section 5.2

**>> Page 18, line 3→ How sensitive is your analyze to the accuracy of the forcing models
in describing the conditions, especially the vertical mixing?**

The choice of turbulence model (usually part of the ocean model) should in deed affect the analysis because the level of turbulence determines how much oil resurfaces, and how fast it resurfaces. This is particularly important because the accuracy of such turbulence schemes to describe diffusion of oil droplets is not well studied. In our simulations, a state of the art 2nd order turbulence scheme is used, but we do not have the means to evaluate it's performance in the present data set. In one of our next experiments we are planning to measure subsurface distribution of oil, which will hopefully allow us to address this question further.

**>> It would be good to emphasize in the conclusions what is general knowledge in the behavior of different oil types and what are your findings.**

The last paragraph is now splitted into two parts, stating which findings confirm established knowledge, and which findings add the the understanding of oil spills. We also state in this section what the respective advantages of the two droplet size spectra are, according to our simulations.

**Other changes**

To account for latest published literature, we include a reference to Callaghan (2018) who provides a parameterization for whitecap area coverage as function of wave energy dissipation (as we earlier have written does not exist). This new paper should have an impact on future oil spill modeling, as we discuss:

[revised manuscript text omitted]